# Coronectomy of Mandibular Third Molar: Four Years of Follow-Up of 130 Cases

**DOI:** 10.3390/medicina56120654

**Published:** 2020-11-27

**Authors:** Saverio Cosola, Young Sam Kim, Young Min Park, Enrica Giammarinaro, Ugo Covani

**Affiliations:** 1Department of Stomatology, Tuscan Stomatologic Institute, Foundation for Dental Clinic, Research and Continuing Education, 55042 Forte dei Marmi, Italy; s.cosola@hotmail.it (S.C.); covani@covani.it (U.C.); 2Gangam Dental Office, Seoul 06614, Korea; doctorkimys@gmail.com (Y.S.K.); min1810@hotmail.com (Y.M.P.); 3Department of Oral and Maxillo-Facial Surgery, Seoul National University, Seoul 06614, Korea

**Keywords:** wisdom tooth, coronectomy, oral surgery, intraoperative complications, mini-invasive

## Abstract

Inferior wisdom teeth extraction surgery may have some complications that, in some cases, could be prevented by a correct diagnosis and minimal surgery. Coronectomy is a technique used for wisdom teeth surgery where only the crown is extracted and the root/roots are left in situ. This procedure may be controversial, but it could limit the common risks of the extraction procedure. Nowadays, the indication and contraindication of this technique are debated, and clinicians normally extract the entire tooth. The following case series includes the data and follow-up radiographs of 130 patients who received a coronectomy, reporting the safety of the procedure. After a mean follow-up period of four years, no complications occurred. A total of 13 patients showed mobile roots but had no complications or symptoms. The roots migrated in a mesial or coronal direction in 31 patients; in 4 cases, they were removed because of patient preference. Coronectomy is a useful oral surgical procedure in certain complicated cases of mandibular wisdom tooth extraction.

## 1. Introduction

In 2000, the UK National Institute for Health and Care Excellence (NICE) published guidelines on wisdom tooth extraction. The NICE guidance recommends that third molars should be extracted only in cases of pathologies such as carious lesions, periapical lesions, recurrent pericoronitis, cystic/neoplastic lesions, and lesions of the second molar [1].

The most common and severe complications of third molar extraction surgery include dry socket, postoperative infection, alveolar bone fracture, oroantral communications, damage of inferior alveolar nerve or lingual nerve, and mandibular fracture in rare cases. Therefore, intentional coronectomy is a well-established technique whereby the root/roots of the wisdom tooth are left in situ and only the crown is sectioned and removed (odontectomy). This procedure has proven to be effective at reducing the risk of mandibular third molar surgery, but it retains its own complications [2].

Recently, several randomized control trials have revealed that the incidence of nerve damage of mandibular third molar extraction is lower in coronectomy compared to complete extraction surgery [3,4]. Nevertheless, modern literature demonstrates that coronectomy of the mandibular third molar is still an unconventional treatment and its indications are not totally clear for clinicians [5].

Intentional coronectomy has been proposed due to emerging evidence of its capacity to reduce several intraoperative risks and to justify, in defensive dentistry, the possibility of non-intentional coronectomy [6].

The indications suggested for coronectomy are [7] the following:Lower wisdom tooth radiographically close to the inferior alveolar canal;Signs of narrowing or diversion (loop) of the inferior alveolar canal;Roots are darkened in the apical third, with the inferior alveolar canal interrupted;Interruption of the lingual cortical bone;Vital tooth without caries, periodontal, or periapical pathology.

Coronectomy is contraindicated in cases of severe infection of wisdom teeth (such as of caries or periapical pathologies), medically compromised patients (especially if they are immunocompromised or under radio- or chemotherapy), and wisdom teeth that can be completely removed with low surgical risk. Moreover, coronectomy is not logical in the case of horizontal teeth because roots could be exposed in the same way as the crown [8].

The procedure described in the literature recommends leaving in position only 5 mm or less of the roots, to extract roots with a periapical acute infection or granuloma, and to remove the entire root in case of mobility [5].

This case series aims to record data and build indication guidelines on coronectomy of the wisdom tooth.

## 2. Methods and Materials

A total of 130 radiographs of 130 patients who underwent coronectomy was extracted from the hospital database of Leon Dental Clinic, Seoul, South Korea (*n* = 110) and Tuscan Stomatological Institute, Viareggio, Italy (*n* = 20). As a prerequisite of minimally invasive wisdom tooth surgery at both the Korean and Italian centers, it was mandatory to understand the correct anatomy and position of the tooth using orthopantomographic (OPT) and cone-beam computed tomography (CBCT) [9]. The surgical technique described by Kim et al. (2018) for wisdom tooth extraction was followed. Kim’s approach is based on multiple sectioning of the tooth in order to remove as little bone as possible, whereas coronectomy became an option after 2015 [10].

The majority of coronectomy cases (*n* = 110) were performed intentionally after 2015, with an average follow-up period of 4 years, while the rest of the coronectomy cases (*n* = 20) were performed non-intentionally between 2010 and 2015, with an average follow-up period of 7 years. The authors must precise that the modern intentionally coronectomy performed is actually “post-intentionally”. It is an intra-operative decision to extract the tooth or to remain intentionally the root/roots according to biological cost-benefit ratio.

### 2.1. Inclusion/Exclusion Criteria

All 130 cases were young adults aged between 24 and 34 years old. Patients younger than 24 years old could not have formed wisdom tooth roots yet; therefore, no data were recorded. None of the included patients had systemic diseases or smoking habits. Immediately before the surgery, all patients in the both center used a mouth rinse of chlorhexidine digluconate solution 0.2% for 1 min and 625 mg of amoxicillin; then, in Korean center after the surgery, they continued the antibiotic therapy for 5 days. All patients in the Italian center received ozone therapy immediately after the surgery; later, they continued the antibiotic therapy for 5 days (875 mg) and corticosteroids for 5 days just if it is necessary.

All patients included in this study consented to the use of their clinical and laboratory findings for academic purposes with the concealment of their personal data. This study was conducted in accordance with the Declaration of Helsinki for human studies and is reported according to the CARE guidelines [11,12].

### 2.2. Statistical Analysis

Descriptive statistics for patients’ characteristics and treatment outcomes were conducted using Microsoft Excel -Office 365 (Microsoft Corp., Redmond, WA, USA, 2020). Student’s *t*-test (*p* < 0.05) was used to analyze the differences between subgroups according to the follow-up, complication, the clinical intervention, and the type of coronectomy (intentional or nonintentional).

## 3. Results

The mean age of included patients was 27.57 ± 3.10 (24–34) years old. Gender was equally distributed over the cohort, with 66 females (50.8%) and 64 males (49.2%). During the follow-up period, which ranged between 3 to 7 years, with a mean value of 4.38 ± 1.39 years, no patient complained of pain or other symptoms in the area where the coronectomy was performed. Anamnestic data are reported in Table 1. On the 11-point visual analog scale (VAS) for patients’ reported pain, the mean reported value was 2.23 ± 1.29 (0–6). Briefly, 9 patients (6.9%) reported “0”, 25 patients (19.2%) reported “1”, 41 patients (31.5%) reported “2”, 33 (25.4%) patients reported “3”, 17 (13.1%) patients reported “4”, and only 2 and 3 patients reported “5” and “6”, respectively.

Only 6 patients out of 130 requested the removal of the roots in another surgery after a variable period of 3–9 months because of fear of future chronic infection; in another 4 patients, the roots were removed because of mesial and coronal migration. In a total of 15 cases out of 130, the roots had partial mobility during the surgery, and a total of 31 roots migrated in a mesial (and coronal) direction during the follow-up period.

Events that occurred after surgery are reported in Table 2.

No statistically significant differences were observed in terms of patients’ reported pain and complications in the Korean or Italian center (*p* > 0.05); there were no differences in terms of complications according to gender, age, or follow-up.

The *t*-test highlighted significant differences in terms of the postsurgical event “mesial migration of the remaining fragment” in a longer follow-up period, which was used to analyze differences between the subgroups according to the follow-up, complication, the clinical intervention, and the type of coronectomy (intentional or nonintentional).

As an example of all these surgical cases, the synopses of two patients are reported according to the CARE guidelines [12].

### 3.1. Case No. 1: Nonintentional Coronectomy

A 30-year-old male patient without systemic diseases had the right mandibular wisdom tooth extracted seven years ago due to malposition, which caused pain and sensitivity to its adjacent second molar in addition to recurrent inflammation in the gingiva around the partially erupted wisdom tooth (Figure 1a). The main clinician (Kim) noticed a dark area on the preoperative x-ray, corresponding to the apical region of the root (classified as Youngsam’ sign), thus implying that there is strict contact between the root and the lingual side of the cortical bone. To perform safe surgery, a CBCT was performed before the surgery.

The position of the apical third of the roots was evaluated using the CBCT; nevertheless, fracture of the root during surgery was not avoided. This intraoperative complication made the clinician decide to perform only coronectomy instead of complete extraction. As an example of nonintentional coronectomy, Figure 1b reveals, during suture thread removal, how the bone healed and covered the remaining region of roots after 10 days.

In Figure 1c, the OPT after 1 year from the extraction showed complete bone healing, with a re-epithelialization of the wound. The OPT was required to check the entire oral cavity and the contralateral wisdom tooth; in correspondence with the right mandibular wisdom tooth, no symptoms were reported by the patient. In Figure 1d, a radiograph after 7 years of follow-up revealed how the bone had healed completely, with no signs of inflammation around the roots. Moreover, the patient did not complain of any pain or sensitivity in the area.

### 3.2. Case No. 2: Intentional Coronectomy

A 25-year-old female patient without systemic diseases had the right mandibular wisdom tooth extracted three years ago due to mesioangulation, which caused pain and sensitivity to its adjacent second molar in addition to recurrent inflammation in the gingiva and pericoronitis (Figure 2a).

The main clinician (Kim) decided to perform a coronectomy to avoid any risk of nerve damage in this young patient who had odontophobia; therefore, the surgical procedure was fast and without bone cuts. Using CBCT, the position of the apical third of the roots was confirmed to be very close to the inferior alveolar nerve.

After 14 days, the patient was recalled for suture thread removal, where she was free of pain (VAS score “2”) and showed no sign of fear while in the dental office (Figure 2b). Three years later, the patient came again to the dental clinic for a dental check-up and for the extraction of the left mandibular wisdom tooth, and she reported no sign of pain or disturbance in the area of the right mandibular wisdom tooth. Figure 2c shows the complete formation of the alveolar cortical bone.

## 4. Discussion

In recent years, the participation of general dentists in oral surgery, implant dentistry, and wisdom tooth extraction has been increasingly possible because of socioeconomic reasons. Furthermore, advances in technology have pushed clinicians to perform complicated cases, even with inadequate experience.

All patients included in the analysis had a normal postoperative follow-up without pain or complications, probably due to the minimally invasive surgical procedures and the postoperative medical care. No differences were observed in terms of patients’ reported pain and complications in the Korean or Italian center; there were also no differences in the other subgroups taken into account (gender, age, follow-up); however, a limitation of this case series is that the dose of antibiotici therapy was slighty different in the Italian patients who also received postoperative ozone therapy. The difference in the pharmacological management of patients is one of the classical biases in retrospective multi-centric case series or studies [13].

Contraindications of coronectomy include the presence of periapical lesions, which are actually very rare in wisdom teeth [14]. Nevertheless, the present authors suggest the evaluation of the risk–benefit ratio if, in some cases, coronectomy is less invasive than the risk of chronic infection. It could be the case of a wisdom tooth with a small periapical lesion that has a great anatomic risk of fracture of the mandible. Despite the length of the remaining root, the authors’ clinical experience indicates that it is more important for the root to be completely surrounded by bone for at least 2 mm; it means that the coronal portion of the cortical bone must be more than 2 mm from the tooth fragment without fragments of enamel [7]. In this case, there were 56 patients with roots longer than 5 mm, such as the second case, but no complications occurred during the follow-up period of 3 years.

Another common contraindication of coronectomy was confuted by these cases as 13 patients with mobile roots that were left inside had no complications during the follow-up; 2 of them with mesial migration needed to extract the teeth. In some cases, after coronectomy, the roots can migrate in either a mesial or coronal direction, and, after several months/years, it will be necessary to perform a second surgery (in case of signs or symptoms or if the fragment is visible upon oral inspection) to remove the remaining roots. When this occurs, usually there is less surgical risk because the root has migrated and the distance from the nerve has improved so the coronectomy procedure must be considered successfully. According to the best available evidence, migration of the remaining root is reported in 14–81% of cases; 31 out of 130 patients in this study reported it, with only 4 cases having to remove it as ossification of the socket had slowed down or prevented root movement [15].

Moreover, the mobility of the roots could bring the teeth to necrosis; some previous studies used mineral trioxide aggregate (MTA) to close the roots left in the bone [16]. The use of MTA was accompanied by abscess or infection of the roots, which did not occur in the present case series, even in the roots with certain mobility. Other case series have reported no case of persistent symptoms attributed to the beheaded wisdom tooth; contrarily, the pulp chambers retained a vital blood supply without evidence of potential infective risk or the need for MTA in recent clinical studies [17,18].

In line with previous results, coronectomy seems to be a safe surgical procedure in most cases, with reduced complications compared to complete wisdom tooth extraction [8,19]. However, the incidence of nerve damage is low in coronectomy; the gold standard is still a complete extraction [20]. Some studies have mentioned that intentional coronectomy is not a simple surgical procedure for beginners as it is a procedure performed by experts to prevent nerve damage [4,21].

Based on the aforementioned evidence, the authors suggest that coronectomy is a valid alternative to complete wisdom tooth extraction, and it can be chosen in certain cases, such as the following [22]: high risk of apical fracture because of thin and curved roots;close proximity of the nerve to the root and the patient presents pain during extraction;close proximity of the root to the lingual plate, as visible in CBCT or OPT, corresponding to an apical radiotranslucency sign;patients with coagulation dysfunction, so that oral surgery must be minimally invasive;intraoperative complications (surgery time, blooding, pain, patients’ discomfort).

The only challenge is to leave the root positioned at least 2 mm below the crestal bone level to avoid dehiscence and reinfection. This surgical procedure should be acceptable from both legal and clinical points of view because it could be useful or necessary in some cases. Despite the fact that additional standardized clinical studies are warranted, this case series, with several limitations (pharmacological therapy, different follow-up periods, different clinicians, different ethnicity), has offered an indication for medico-legal issues because leaving the root inside, in some cases, is a reasonable and justifiable procedure.

## 5. Conclusions

Intentional coronectomy and nonprogrammed coronectomy are valid oral surgical procedures that may help the clinician in certain cases of mandibular wisdom tooth extraction. The long follow-up period of these cases has revealed that coronectomy is probably a safe procedure, and the removal of remaining roots is required in around 5% of cases due to the mesial migration of the fragment and not any symptoms or reinfection. Longer evaluation of these cases should be performed, and other clinical studies to evaluate the patients’ comfort and the safety of the surgery need to be conducted.

## Figures and Tables

**Figure 1 medicina-56-00654-f001:**
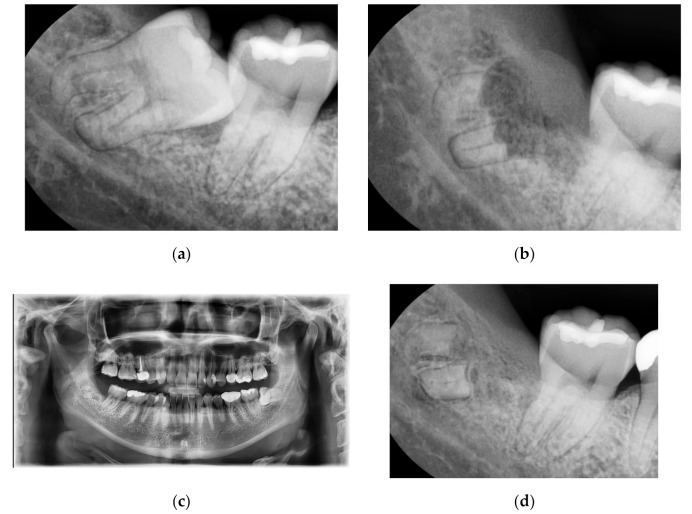
(**a**): Right mandibular wisdom with pain and sensivity due to malposition. (**b**): X-ray after 10 days from the coronectomy. (**c**): OPT after one year; (**d**): X-ray after a radiograph after seven years of follow-up.

**Figure 2 medicina-56-00654-f002:**
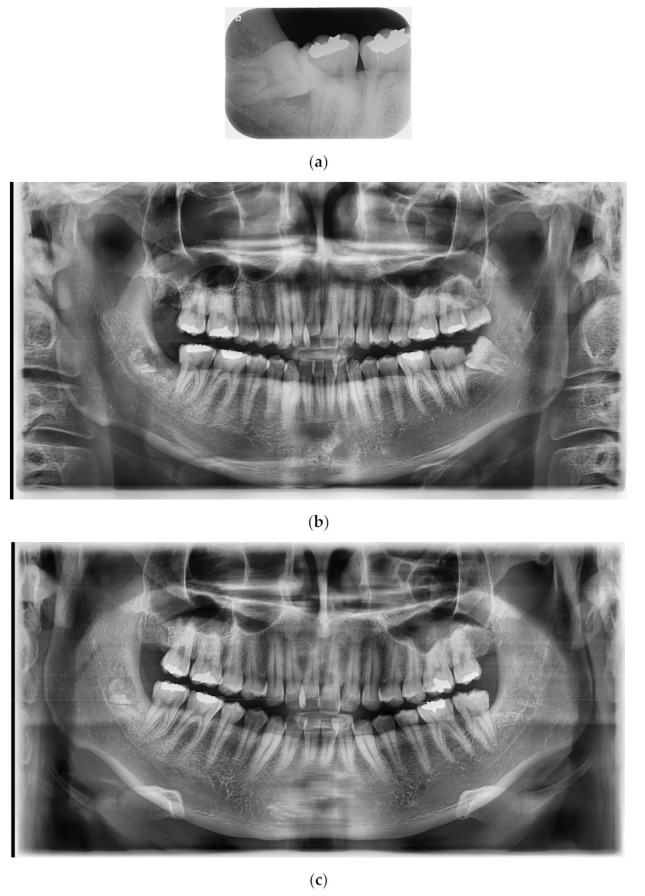
(**a**): Right mandibular wisdom with pain and sensivity due to pericoronitis. (**b**): X-ray after 14 days from the coronectomy. (**c**): OPT after three years of follow-up.

**Table 1 medicina-56-00654-t001:** Anamnestic data and visual analog scale (VAS) score of the sample.

		Age	Gender	Follow-Up	VAS
N	Valid	130	130	130	130
	Missing	0	0	0	0
Mean		27.54	0.49	4.38	2.32
Std. Deviation		3.165	0.502	1.394	1.289
Minimum		20	0	3	0
Maximum		34	1	7	6

**Table 2 medicina-56-00654-t002:** Complications and mean follow-up period for each event occurred.

Events	Coronectomy (tot. 130)	Mean Follow-Up
Extraction of remanent fragment	6	6.43 ± 2.44 months
Mobility of the fragment during surgery	15	4.6 ± 1.35 years
Mesial migration of the fragment	31	5.23 ± 1.75 years
Extraction of remanent fragment	4	6.5 ± 0.58 years
Severe complications	0	/

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
