# Peer review of "Coronectomy of Mandibular Third Molar: Four Years of Follow-Up of 130 Cases"

_medicina, 2020, doi:10.3390/medicina56120654_

Round 1

Reviewer 1 Report

-please report inclusion/exclusion criteria in methods section.

-add a statistical analysis paragraph. Improve statistical tests and explore more viariables and associations. 

-add study limitation paragraph, because of non-control aim of study about the role of antibiotics and corticosteroids

-" In a total of 15 cases out of 130, the roots had a" -> complete the sentence.

-report a table of complications and mean-time follow-up for the occurence

-Line 146-152 is not supported by results. Please rephrase and discuss according results.

-Most of results are presented in discussion, please move them to results

-Line 159-160 rephrase. I can't understand what do you mean.

-

Author Response

Dear reviewer thank you very much to have revised our case series.

It means a lot for me and I tried as much as possible to revise the paper according to your precious advices. We reported the inclusion/exclusion criteria, even if it is a case series and also the statistical analysis paragraph improved. We explored the statistical variable of follow-up and complication occurred in table 2. Improve statistical tests and explore more viariables and associations. I talked about the several limitation of the paper.

We completed the sentence "In a total of 15 cases out of 130, the roots had a" and corrected other senteces in results and discussion.

Best regards,

Dr. Saverio Cosola

Reviewer 2 Report

Your paper is a great contribution, useful for the students of post-graduate program in oral surgery.

The research is well carried out.

Regarding antibiotic therapy protocol, it would be useful to cite guidelines and some review, like the following:

Cervino, G.; Cicciù, M.; Biondi, A.; Bocchieri, S.; Herford, A.S.; Laino, L.; Fiorillo, L. Antibiotic Prophylaxis on Third Molar Extraction: Systematic Review of Recent Data. Antibiotics 2019, 8, 53.

in order to better clarify this aspect.

Could you add other clinical tips and implications? they are wothwhile in a case-report article.
Could you add a table? datas would be more readable.

Author Response

Dear reviewer thank you very much to have revised our case series.

We modified the paper according to your precious advices, adding references and tables.

Best regards,

Dr. Saverio Cosola
